# The histologic phenotype of lung cancers is associated with transcriptomic features rather than genomic characteristics

Ming Tang[1,11], Hussein A. Abbas [2,11], Marcelo V. Negrao [3], Maheshwari Ramineni[4], Xin Hu[1], Shawna Marie Hubert [3], Junya Fujimoto[5], Alexandre Reuben [3], Susan Varghese[3], Jianhua Zhang [1], Jun Li[1], Chi-Wan Chow[5], Xizeng Mao[1], Xingzhi Song[1], Won-Chul Lee[1], Jia Wu [6], Latasha Little[1], Curtis Gumbs[1], Carmen Behrens[3], Cesar Moran [7], Annikka Weissferdt[7], J. Jack Lee [8], Boris Sepesi[9], Stephen Swisher [9], Chao Cheng [10], Jonathan Kurie [3], Don Gibbons [3], John V. Heymach [3], Ignacio I. Wistuba[3,5], P. Andrew Futreal [1✉], Neda Kalhor [7✉] & Jianjun Zhang [1,3✉]

Histology plays an essential role in therapeutic decision-making for lung cancer patients. However, the molecular determinants of lung cancer histology are largely unknown. We conduct whole-exome sequencing and microarray profiling on 19 micro-dissected tumor regions of different histologic subtypes from 9 patients with lung cancers of mixed histology. A median of 68.9% of point mutations and 83% of copy number aberrations are shared between different histologic components within the same tumors. Furthermore, different histologic components within the tumors demonstrate similar subclonal architecture. On the other hand, transcriptomic profiling reveals shared pathways between the same histologic subtypes from different patients, which is supported by the analyses of the transcriptomic data from 141 cell lines and 343 lung cancers of different histologic subtypes. These data derived from mixed histologic subtypes in the setting of identical genetic background and exposure history support that the histologic fate of lung cancer cells is associated with transcriptomic features rather than the genomic profiles in most tumors.

[1] Department of Genomic Medicine, Division of Cancer Medicine, The University of Texas MD Anderson Cancer Center, Houston, TX 77030, USA. [2] Medical Oncology Fellowship, Division of Cancer Medicine, The University of Texas MD Anderson Cancer Center, Houston, TX 77030, USA. [3] Department of Thoracic/Head and Neck Medical Oncology, Division of Cancer Medicine, The University of Texas MD Anderson Cancer Center, Houston, TX 77030, USA. [4] Department of Pathology, Baylor College of Medicine, Houston, TX 77030, USA. [5] Department of Translational Molecular Pathology, Division of Pathology and Laboratory Medicine, The University of Texas MD Anderson Cancer Center, Houston, TX 77030, USA. [6] Department of Imaging Physics, Division of Diagnostic Imaging, The University of Texas MD Anderson Cancer Center, Houston, TX 77030, USA. [7] Department of Pathology, Division of Pathology and Laboratory Medicine, The University of Texas MD Anderson Cancer Center, Houston, TX 77030, USA. [8] Department of Biostatistics, Division of Basic Sciences, The University of Texas MD Anderson Cancer Center, Houston, TX 77030, USA. [9] Department of Thoracic Surgery, Division of Surgery, The University of Texas MD Anderson Cancer Center, Houston, TX 77030, USA. [10] Institute for Clinical and Translational Research, Baylor College of Medicine, Houston, TX 77030, USA. [11]These authors contributed equally: Ming Tang, Hussein A. Abbas. ✉email: AFutreal@mdanderson.org; nkalhor@mdanderson.org; jzhang20@mdanderson.org

Lung cancer is the leading cause of cancer death in the United States with an estimated 1,898,160 new cases and 608,570 deaths expected in 2021[1]. Histopathology continues to play an essential role in prognosis and choosing appropriate treatment[2]. Largely determined by morphology, primary lung cancers are histologically classified into small cell lung cancers (SCLC) and non-small cell lung cancers (NSCLC), with the latter including adenocarcinoma (LUAD), squamous cell carcinoma (LUSC), and large-cell neuroendocrine carcinoma (LCNEC) as the main histologic subtypes. However, consensus histologic confirmation can sometimes be challenging and therefore impacts optimal treatment choices[3,4]. The molecular mechanisms determining the tumor histology are unknown. Previous studies revealed that tumors from different patients or even multiple independent primary lung cancers within the same patients can have identical morphology yet share no mutations[5], while there can be a morphologic difference in different regions within the same tumors that share the majority of mutations[6]. These findings suggest that morphology may not be primarily determined by genomic features.

About 5% of primary lung cancers can present with a mixed histologic pattern, where multiple distinct histologic components present within the same tumors, often referred to as combined or mixed histology[7,8]. Tumors with mixed histology provide a unique opportunity to study the molecular basis for histology determination as different histologic components share the same genetic backgrounds and exposure history. There have been a few studies on lung cancers of mixed histology, most of which focused on the genomic changes of adenosquamous lung cancers. The majority of these studies revealed shared driver mutations between different histologic components[8–14]. These findings are overall in line with the prior hypothesis that genomic changes were not the main determinants of histology. However, these studies only covered hotspot driver mutations or small gene panels, while mutations of other genes with essential biological functions and other genomic alterations such as somatic copy number alterations (SCNA) were not investigated. Thus, the relationship between genomic alterations and histology was not fully addressed.

In the current study, we leverage three unique datasets to show that the histologic phenotype of lung cancers is associated with transcriptomic features rather than genomic characteristics: (1) whole-exome sequencing (WES) and transcriptomic data from 19 microdissected tumor regions of different histology from 9 primary lung cancer patients with mixed histologic patterns including 6 LUAD, 6 LCNEC, 3 SCLC, 3 LUSC, and one poorly differentiated NSCLC-NOS; (2) transcriptomic data from 141 cell lines of different histologic subtypes from the Cancer Cell Line Encyclopedia (CCLE)[15] including 14 LCNEC, 57 LUAD, 48 SCLC, and 22 LUSC; (3) transcriptomic data from a total of 343 patients including 14 LCNEC, 273 LUAD, 9 SCLC, and 47 LUSC with lung cancers of different histologic subtypes[16,17].

## Results

**Patient characteristics.** The clinicopathologic characteristics of the nine patients with lung cancers of mixed histology are summarized in Supplementary Data 1. The median age at diagnosis with lung cancer was 67 years (range 47–79 years). All patients were current (3/9) or former (6/9) smokers. Eight patients had two distinct histologic subtypes, while one patient had three different histologic components (Supplementary Data 1). Representative images of hematoxylin and eosin and immunohistochemical (IHC) staining of these tumors are shown in Supplementary Figs. 1a–d and 2a–d, respectively. Different histologic components of each tumor of mixed histology were manually microdissected, which resulted in 19 different tumor tissues including 6 LUAD, 6 LCNEC, 3 SCLC, 3 LUSC, and 1

poorly differentiated NSCLC-NOS that were subjected to WES and microarray RNA profiling. The most common combination of mixed histology was LCNEC-LUAD in 4/9 patients, followed by LCNEC-LUSC and SCLC-LUAD subtypes in 2/9 patients each, and 1 patient had SCLC-LUSC subtypes.

**Shared mutations across different patients and distinct histologic subtypes.** We first investigated whether the mutations overlapped between different histologic components within the same tumors and whether there were particular mutations shared across the same histologic components from different patients. Overall, different histologic components from the same tumors shared the majority of mutations (Fig. 1, Supplementary Data 2, and Supplementary Fig. 3a–i). The percentage of shared mutations within the same tumors ranged from 12.1% to 98.4% with a median of 68.9%, similar to that between different regions within the same tumors of the same histology[6] (68.9% vs 72%, $p = 0.46$, Wilcoxon rank-sum two-side test). These results are consistent with previous findings from adenosquamous mixed histology lung cancers[7–11], suggesting somatic mutations may not be the primary determinants of histology in most tumors. Of note, in Pa35, only 12.1% of mutations were shared between the SCLC and LUAD components. Therefore, we cannot exclude the contribution of genetic alterations in histologic determination in a subset of tumors.

**Similar mutational processes are occurring between different histologic components within the same tumors.** It is well known that different cancer types have distinct mutational signatures[18] suggesting different mutational processes in play reflecting different genetic backgrounds and exposure etiologies associated with different cancer types. To understand whether the mutational processes are histology-specific in these lung tumors of mixed histology in the context of identical genetic background and exposure history, we calculated the mutational spectrum and mutational signatures in each histologic component. Overall, a similar mutational spectrum was observed between different histologic components within the same tumors (Fig. 2a). We next calculated the contribution of 30 signatures of mutational processes in cancer[18] (Fig. 2b, c). Not surprisingly, Signature 4 (associated with smoking and tobacco carcinogenesis) was the most dominant in seven of nine patients consistent with their smoking history (Fig. 2c). Two exceptions were patients Pa35 and Pa26, who were both former light smokers with a 2.5 and 5 pack-year smoking history, respectively, and both quit >20 years ago. Other common signatures in this cohort of tumors included Signature 1 (associated with spontaneous deamination of 5-methylcytosine), Signatures 2 and 13 (associated with APOBEC-mediated mutagenesis), and Signature 16 (etiology-unknown). Similar to the mutation spectrum, the mutational signatures were also overall similar between different histologic components within the same tumors, while none of the mutational signatures enriched in certain histologic components were shared across different patients. Taken together, these data suggest that mutational processes were not histology-specific, but rather patient-specific, likely determined by the particular exposure history and host factors in each patient.

**Somatic copy number aberration profiles are similar between different histologic components within the same tumors.** SCNA is another key feature of human malignancies that could potentially impact the expression of large groups of genes. We next delineated the genome-wide SCNA profiles. As shown in Fig. 3a, b, the overall SCNA profiles were similar between different histologic components within the same patients, while drastically different among different patients. Furthermore, we quantified SCNA events using a gene-based SCNA analysis

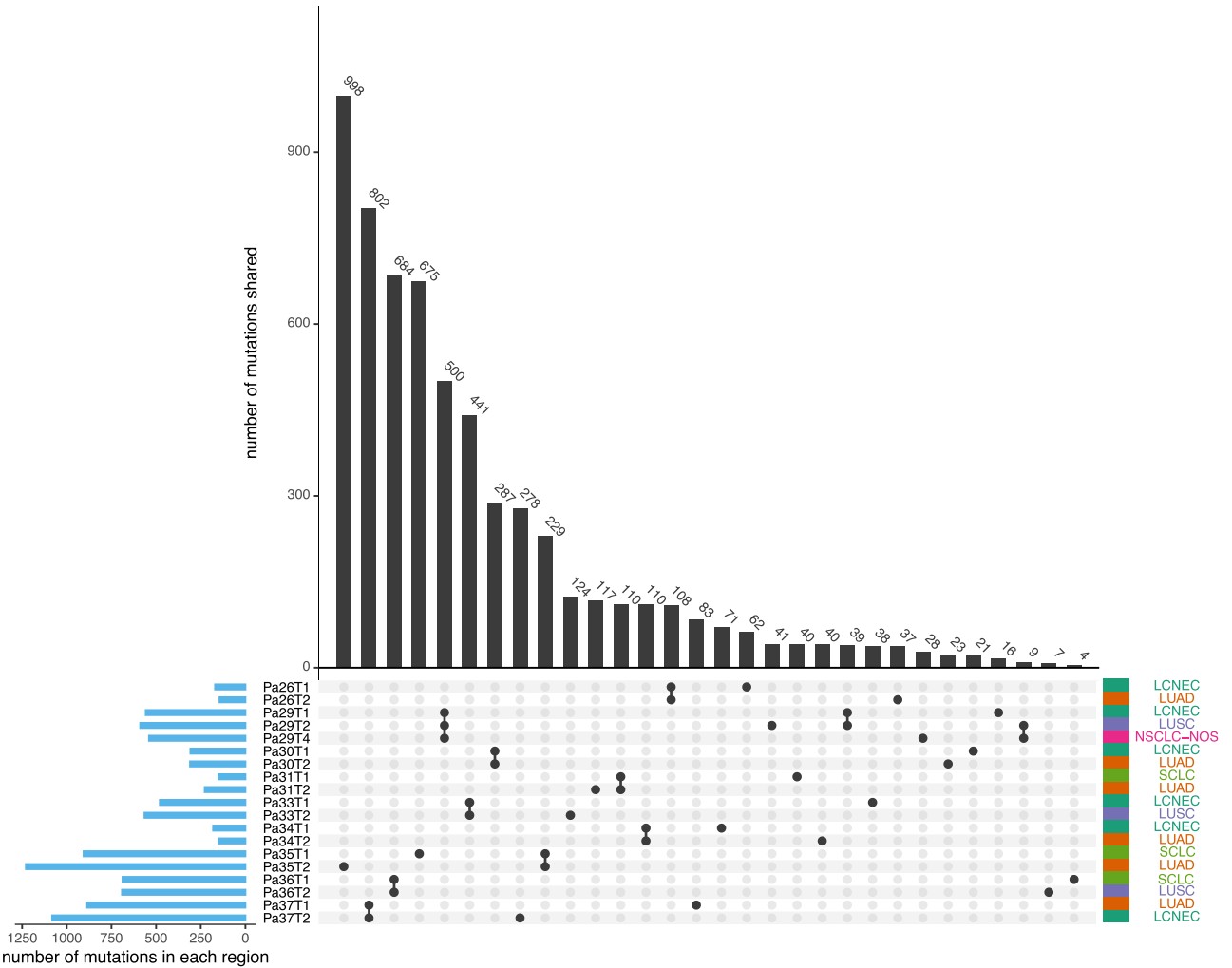

**Fig. 1 Overlapping number of somatic mutations across the samples.** The upset plot demonstrates the shared mutations across samples. Blue bars in the *y*-axis represent the total number of mutations in each sample. Black bars in the *x*-axis represent the number of mutations shared across samples connected by the black dots in the body of the plot. Source data are provided as a Source Data file.

algorithm[19] for exome sequencing data that allows comparing the SCNAs between different samples to identify shared and unique SCNA events between different histologic components within the same tumors. To minimize the impact of tumor purity on SCNA analysis, we obtained purity-adjusted log2 copy number ratios for each tumor in this study (see Methods for details). On average, 83% of SCNA events (ranging from 54.7% to 99.1%) were shared between different histologic components within the same tumors suggesting the majority of SCNA events were early molecular events before the separation of different histologic components. No particular SCNAs were found to be enriched in certain histologic subtypes. Furthermore, compared to the intratumor heterogeneity dataset from the TRACERx study[20], at the gene level, the extent of shared SCNA landscape between different histologic components was comparable to that between spatially separated tumor regions within the same NSCLC tumors of the same histology (83% in mixed histology cohort vs 72% in TRACERx cohort, $p = 0.25$, Wilcoxon rank-sum two-side test).

**Similar subclonal architecture between different histologic components.** We next inferred cancer cell fractions (CCF) of all somatic mutations using PyClone[21] adjusting for copy number

changes and tumor purity to determine the subclonal architecture in each histologic component. Overall, the subclonal architecture was similar between different histologic components within the same tumors. Particularly, Pa29 and Pa36 have the CCFs lined up almost on the diagonal line indicating nearly identical subclonal architecture between different histologic components within the same tumors. A substantial proportion of clonal mutations[22,23] were shared across different histologic components of the same tumors and only a small proportion of clonal mutations were private (Fig. 4a–k). Specifically, among the shared mutations, an average of 54.6% (ranging 16–96.5%) were clonal, while only 10.7% (ranging 0.14–35.8%) of private mutations were clonal. One plausible explanation is that the separation of different histologic clones was molecularly late events during the evolution of most tumors when the subclonal architecture was already determined, and no major genomic evolution has occurred after the separation of different subclones giving rise to different histologic components.

**The majority of cancer gene alterations occurred before the divergence of different histologic components of the same tumors.** Cancer gene mutations are known to determine distinct molecular subsets of lung cancers with unique clinical

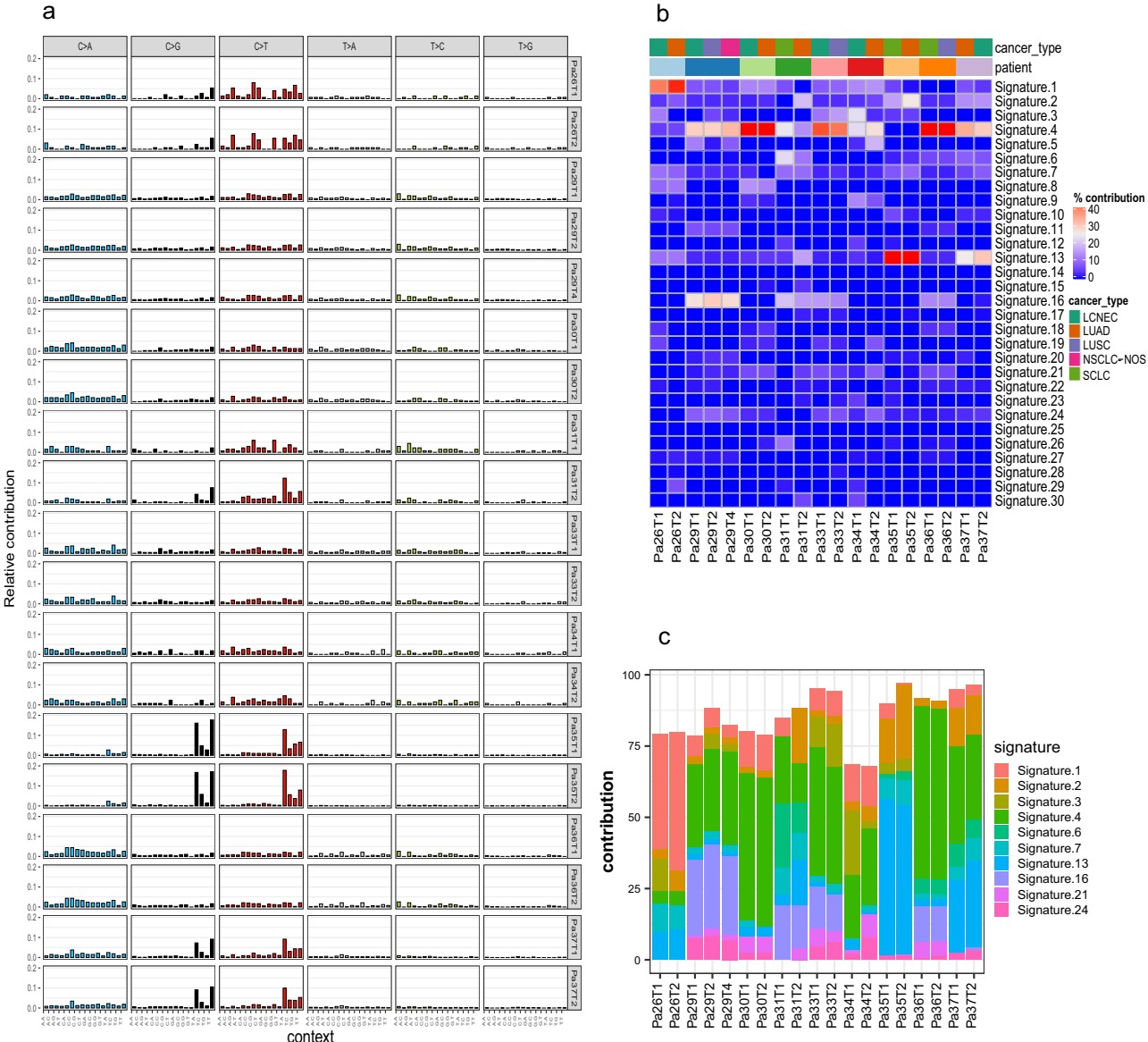

**Fig. 2 Mutational spectrums and signatures are similar across different histologic components within the same patient. a** Bar plots represent the mutational spectrum decomposed by trinucleotide context. **b** Heatmap of the contribution of the 30 COSMIC mutation signatures in each sample. **c** Stacked barplot for the contribution of the top 10 mutation signatures in each sample. Source data are provided as a Source Data file.

presentation and cancer biology and certain cancer gene mutations are even considered pathognomonic for certain histologic subtypes[24]. Among lung cancers, for example, alterations in *EGFR*, *KRAS*, *SMARCA4*, *STK11*, and *KEAP1* are almost exclusively observed in LUADs[24]; LUSCs often carry mutations in *TP53*, *CDKN2A*, *RB1*, *NFE2L2*, *KEAP1*, *PIK3CA*, and *PTEN*[25], while *RB1* and *TP53* are frequently altered in neuroendocrine carcinomas (NEC) including LCNEC and SCLC[26]. We next investigated whether specific cancer gene mutations could determine different histologic patterns in these tumors of mixed histology. A total of 34 canonical cancer gene mutations, defined as nonsynonymous mutations identical to those previously reported in oncogenes[27,28] or truncating mutations in known tumor suppressor genes (TSG), were identified in these 19 specimens (Supplementary Data 3). Importantly, 30 of the 34 canonical cancer gene mutations were clonal in each histologic component (Supplementary Data 3). Furthermore, 31 of the 34 cancer gene mutations were shared between different histologic components within the same tumors (Supplementary Data 3).

To further delineate the evolution of these tumors of mixed histology and understand the timing of cancer gene mutations, we constructed phylogenetic trees and mapped the canonical cancer gene mutations to the trunks (representing early clonal events before separation of different histologic subclones) and branches (representing later subclonal events after separation of cancer cell subclones that gave rise to different histologic components). As demonstrated in Supplementary Fig. 4, the majority of canonical cancer gene mutations were early trunk events before the divergence of different histologic subclones. Interestingly, in patient Pa35, a *PIK3CA* p.M1043I mutation was shared between the SCLC and LUAD components, while a *PIK3CA* p.E542K was only detected in the LUAD component (Supplementary Fig. 4g and Supplementary Data 3). Similarly, in Pa37, a *PIK3CA* p.E545K was identified in both LUAD and LCNEC components, while a *PIK3CA* p.H1047R was private to the LUAD component (Supplementary Fig. 4i and Supplementary Data 3). These findings are reminiscent of heterogeneity studies in kidney[29] and lung cancers[5,20,30], where different mutations in the same

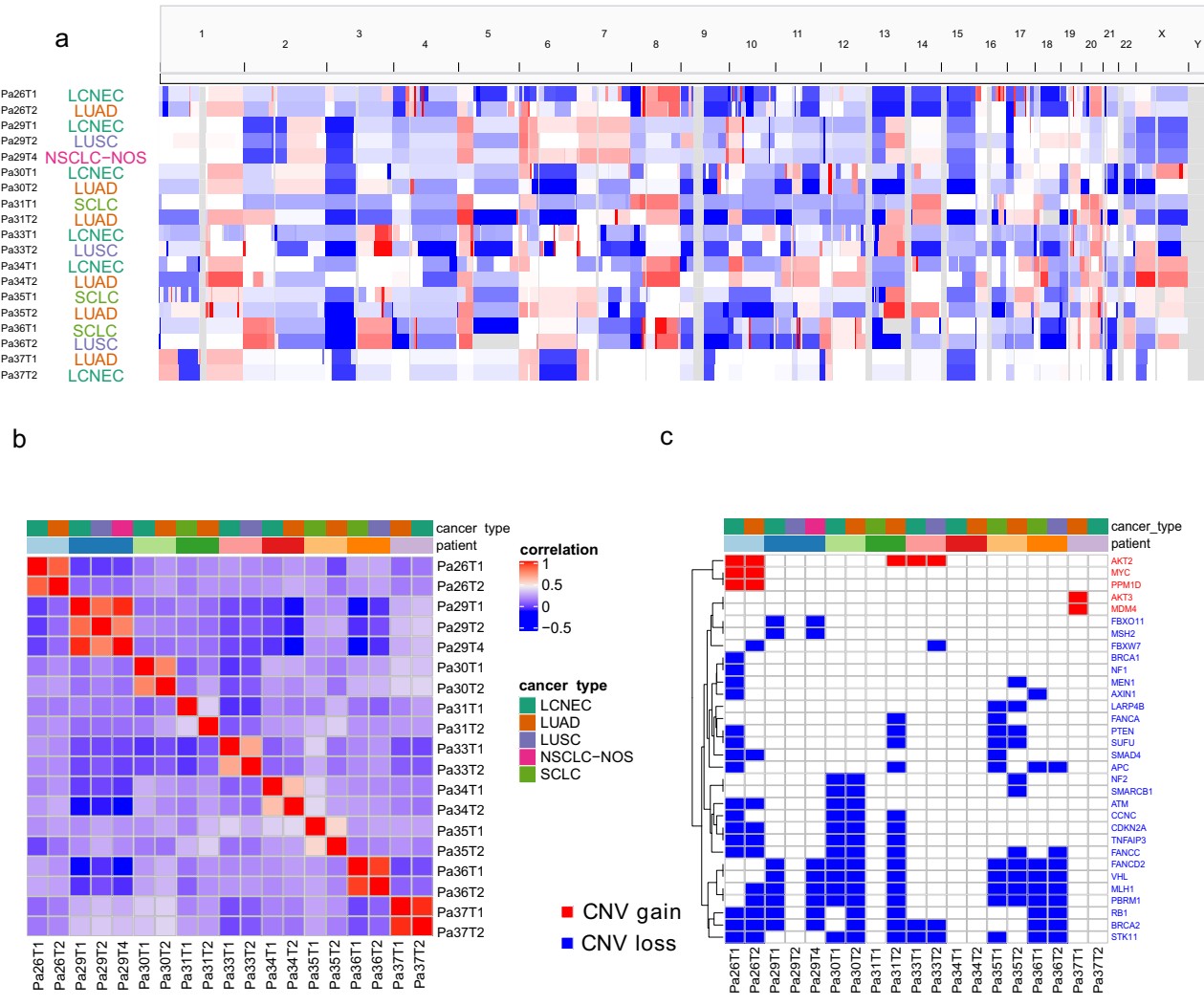

**Fig. 3 Somatic copy number aberration (SCNA) analysis demonstrated similar copy number changes between different histologic components within the same patient. a** IGV screenshot of genome-wide SCNA profile for each sample. **b** Heatmap of the correlation of SCNA at the gene level. **c** Heatmap of copy number changes from canonical cancer genes of the COSMIC database. Source data are provided as a Source Data file.

cancer genes were identified in different regions within the same tumors or different independent primary tumors within the same patients. These results imply convergent evolution and that even with an identical genetic background and environmental exposure, the evolution of different cancer cell subclones can be driven by distinct molecular events, with possible genetic constraints around certain genes or pathways (*PIK3CA* in case of patient Pa35 and Pa37) that are pivotal for cancer evolution.

Next, we estimated copy number gains of oncogenes and copy losses of TSG based on the COSMIC database[27] in this cohort of tumors of mixed histology (Fig. 3c). A total of 11 copy number gains of 5 oncogenes and 129 copy number losses of 27 TSGs were detected in this cohort of tumors of mixed histology. Similar to cancer gene point mutations, 53.8% of SCNA in oncogenes and TSGs were shared within the same patients. Furthermore, loss of heterozygosity (LOH) of *RB1* was identified in seven out of nine tumors of mixed histology (Supplementary Data 4), in line with that all tumors have NEC components. Importantly, LOH of *RB1* was shared between different histologic components in all seven tumors. These data suggested that the cancer gene mutations and copy number changes were early molecular events acquired before the divergence of different histologic subtypes and maybe

not the major mechanisms determining the histologic fate of cancer cells in lung cancers of mixed histology.

**Specific transcriptomic patterns may be associated with specific histologic subtypes.** As the histology of these lung cancers did not appear to be determined by genomic aberrations, we next sought to explore whether the cell fate is determined at the transcriptomic level. We first performed gene expression profiling of the same tumor regions of distinct histologic subtypes to investigate whether transcriptomic signatures could differentiate histological subtypes. By principal component analysis, the normal lung tissues were separated from the tumor samples highlighting the distinct transcriptomic changes associated with malignant cells (Fig. 5a). Tumor specimens of different histologic subtypes from the same patients overall clustered together, although there was a small cluster of LUAD samples from different patients clustered close to each other (Fig. 5a). In unsupervised hierarchical clustering, different histologic components within the same tumors also tended to cluster together highlighting substantial inter-patient heterogeneity. On the other hand, 8 of the 19 specimens were clustered with specimens from a different patient, significantly more common than that of different tumor regions within the same tumors of same histology, where 2 of

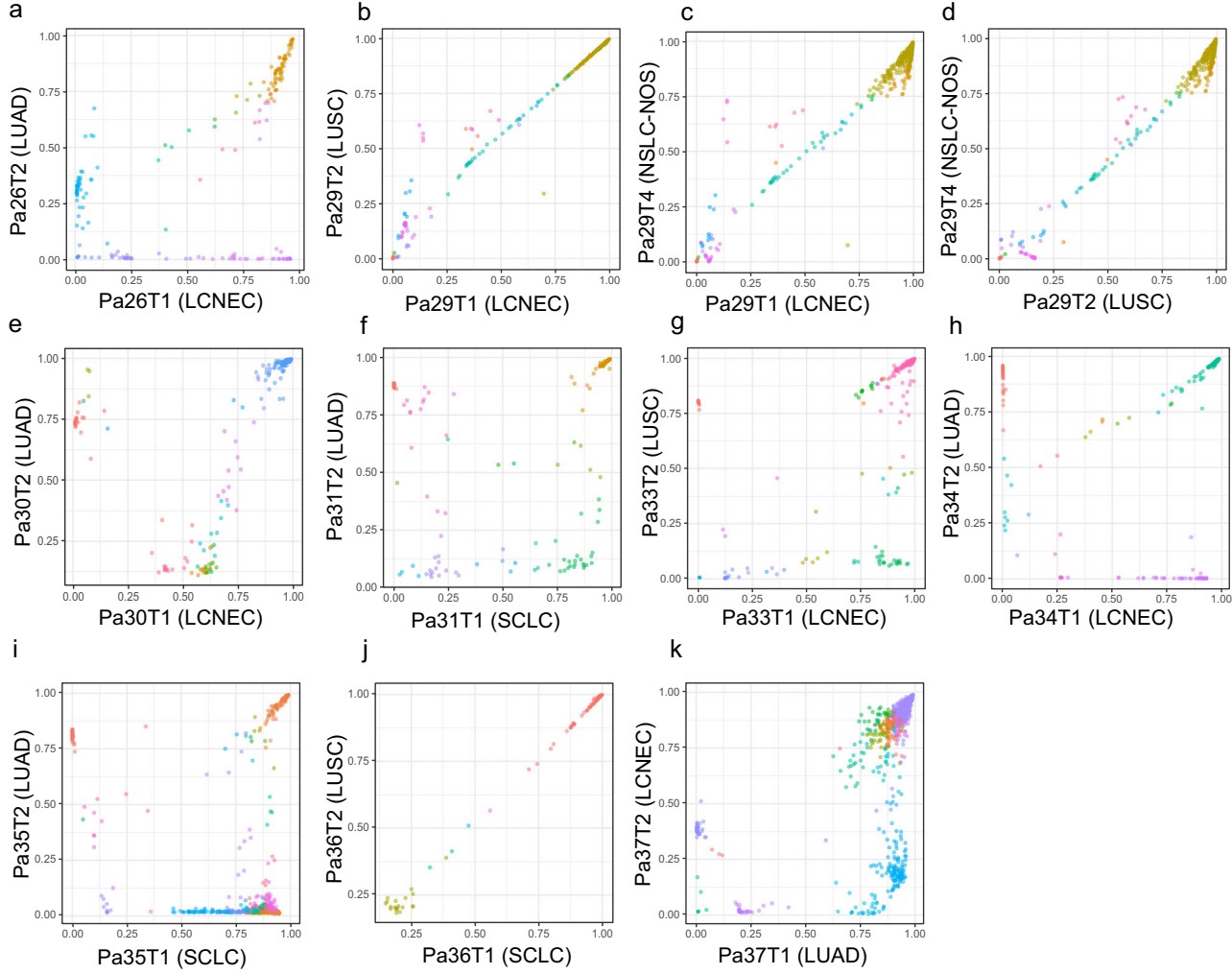

**Fig. 4 Clonality analysis revealed shared clonal mutations between different histologic components within the same patients. a–k** Scatter plots of the cellular prevalence of somatic mutations calculated by PyClone for the two histological components within the same patient. Mutations were clustered by PyClone and mutations of the same cluster were labeled with the same color. Source data are provided as a Source Data file.

35 specimens were clustered with a different patient ($p = 0.001$ by $\chi^2$ test)[31]. Among these eight specimens, four LUAD specimens (Pa26T2, Pa30T2, Pa31T2, and Pa37T1) were clustered together, while Pa35T1 (LCNEC) clustered with Pa37T2 (SCLC) (although Pa35T1 is closer to Pa35T2) and P30T1 (LCNEC) clustered with P31T1 (SCLC) (Fig. 5b)—both LCNEC and SCLC are considered as NEC sharing many biological and clinical features[32]. Similarly, the LCNEC components of patients Pa26 and Pa29 were clustered together. Taken together, these data suggested that in the background of patient-specific gene expression profiles, there may be histology-specific transcriptomic features, associated with distinct histological phenotypes.

**Histology-specific pathways shared with independent cohorts.** To further understand the transcriptomic features associated with different histologies, we evaluated if any Hallmark pathways[33] were enriched in different histologic subtypes. To identify histology-specific pathways, we looked specifically at overlapping pathways in the histologic comparison pairs in different patients that had the same direction of enrichment (either positive or negative). The most concordant pattern was noted in Pa31 and Pa35 with SCLC versus LUAD, whereas three pathways were upregulated and nine pathways were downregulated in SCLC components compared to LUAD components (Fig. 5c).

Interestingly, the three upregulated pathways in SCLC (E2F_Target, G2M_checkpoint, and MYC_target) were associated with cell proliferation, while six of the nine downregulated pathways in SCLC components (IL2, complement, INFG, INFA, TNFA, and inflammatory response) were associated with inflammatory/immune response. In the LCNEC versus LUAD comparisons, there were no pathways with consistent enrichment in all four patients (Fig. 5d). However, compared to LUAD, MYC, G2M, and E2F pathways were upregulated in LCNEC components in 3/4, 3/4, and 2/4 and patients, respectively, while interferon-alpha and interferon-gamma responses were downregulated in LCNEC components in 2/4 and 2/4 patients, respectively (Supplementary Data 5).

To validate these findings, we analyzed the transcriptomic data from another three different cohorts: two previously published large cohorts of primary lung cancers by Karlsson et al.[17], which encompassed 126 primary lung cancers (83 LUAD, 26 LUSC, 3 SCLC, and 14 LCNEC) and by Bhattacharjee et al.[16] with 217 lung cancer patients (190 LUAD, 21 LUSC, and 6 SCLC), as well as 141 cell lines (57 LUAD, 22 LUSC, 48 SCLC, and 14 LCNEC) from CCLE database[15]. Using the same approach for data from tumors of mixed histology, we identified enriched pathways by comparing LCNEC versus LUAD, LCNEC versus LUSC, SCLC versus LUAD, and SCLC versus LUSC of each cohort respectively (Supplementary

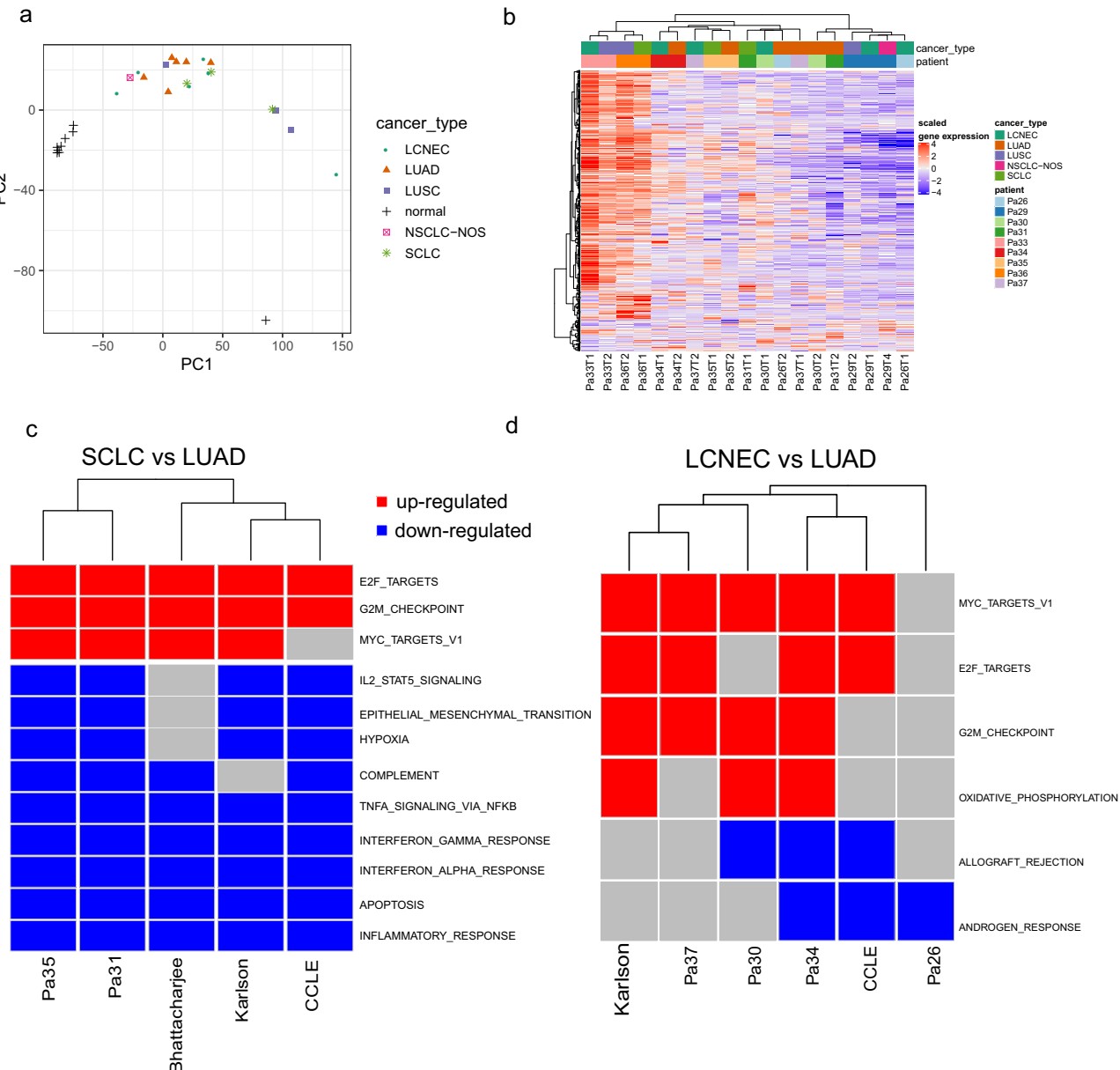

**Fig. 5 Gene expression profile revealed some extent of similarity of the same histologic components across different patients. a** Principal component analysis (PCA) of all histologic components based on gene expression data. **b** Heatmap of the top 500 most variable genes across the samples clustered by both genes and the samples. **c** Commonly upregulated and downregulated pathways comparing SCLC with LUAD across public datasets and in-house dataset. **d** Commonly upregulated and downregulated pathways comparing LCNEC with LUAD across public datasets and in-house dataset. Source data are provided as a Source Data file.

Data 5). We next focused on the pathways that were (1) identified in at least two patients from our mixed histology cohort and (2) validated by at least two of the three datasets (Karlsson cohort, Bhattacharjee cohort, and CCLE). With these criteria, SCLC versus LUAD comparison demonstrated the most consistent pattern with cell proliferation-related pathways upregulated and inflammatory/ immune response pathways downregulated in SCLC ($p$ adj < 0.05) (Fig. 5c). Also, for LCNEC versus LUAD histology pathway analysis, there was significant positive enrichment for cell cycle G/M cell cycle checkpoint and MYC targets for Pa30, Pa34, Pa37, and in the Karlson dataset ($p$ adj < 0.05) (Fig. 5d).

## Discussion

In the immuno-oncology era, histological subtype continues to play essential roles in determining the optimal treatment for lung

cancer patients[34–36]. For example, surgical resection is the main treatment modality for localized NSCLC, while SCLC is usually treated with chemotherapy and radiation even at the localized stage[37]. In the metastatic setting, the chemotherapy regimens are also different for different histologic subtypes. Currently, the mechanisms underlying histologic cell fate are unknown. Understanding the molecular determinants of histology may provide insights to understand the different responses to various treatment regimens and to more effectively leverage histology to guide lung cancer management. Although large-scale studies such as in TCGA have demonstrated that genomic features are largely distinct between different lung cancer histologic subtypes[24,25,38], genomic alterations do not always agree with histologic subtypes. Targetable genomic alterations such as *EGFR* mutations and *ALK/ROS1* translocations that are pathognomonic for LUAD

have been reported in some LUSC patients and SCLC patients[39,40], suggesting that the histology is not primarily determined by genomic features. However, these analyses are complicated by the distinct genetic background and exposure history in different cancer patients.

Cancers of mixed histology provide a unique opportunity to identify the molecular features associated with different histologic components in the setting of identical genetic background and exposure history. Among the lung tumors of mixed histology, the adenosquamous carcinoma is the most common and most frequently studied subtype while other mixed histology subtypes were rarely investigated. In the current study, we specifically focused on non-adenosquamous lung cancers of mixed histology, particularly tumors with high-grade NEC component including LCNECs and SCLCs. We chose high-grade NECs because they are very different from other lung cancer subtypes and are associated with aggressive cancer biology and poor clinical outcome. We applied WES and gene expression microarray with the intent to depict the comprehensive molecular basis of histology. Analysis of WES data from nine patients with mixed histology demonstrated that different histological components within the same tumors shared a large proportion of identical point mutations, which is consistent with previous studies in adenosquamous subtypes by cancer gene panel sequencing[7–11]. In addition to more comprehensive point mutation data, WES also allowed us to compare different histologic components regarding the SCNA profiles, which demonstrated that different histologic components from the same tumors share the majority of SCNA events. In addition, different histologic components from the same tumors also demonstrated overall similar subclonal architecture and canonical cancer gene alterations. It has been reported that 1–4% of EGFR-mutant LUADs may transform into SCLCs as one important mechanism underlying drug resistance to EGFR tyrosine kinase inhibitor treatment and transformed SCLCs share similar genomic profiles of their parental LUADs[41,42]. For example, Lee et al. showed that transformed SCLCs share a common clonal origin with their parental LUADs and complete inactivation of both RB1 and TP53, a genomic hallmark for SCLC, was observed in the original LUADs[42]. Similarly, Niederst et al.[41] also demonstrated RB1 loss in 100% of transformed SCLCs as well as the original EGFR-mutant LUADs. Taken together, these data suggest that different histologic components were derived from the same progenitor cells and that in most tumors of mixed histology, the divergence of distinct histologic components was a relatively late molecular event conferring inter-histologic heterogeneity and the histologic subtype was not primarily determined by genomic alterations.

There is ample evidence that gene expression profiling can inform lung cancer histology[16,17,43]. Our transcriptomic profiling from histologic subtypes in tumors of the same patient allowed decoupling of the effect of the patient's genetic background and exposures in influencing the transcriptomic signatures. Unlike the similar genomic landscape between different histologic components, intratumor heterogeneity of transcriptomic profiles between different histologic components was significantly higher than spatially separated regions from tumors of the same histology. A substantial proportion of tumor regions clustered more closely together with tumor regions of the same histology from different patients, significantly more common than that in different tumor regions of the same histology[31]. Pathway analysis demonstrated common pathways between different histologic components across different patients, which were further supported by integrative analysis from cell lines and larger cohorts of patient datasets. These were mostly accentuated between SCLC and LUAD as well as LCNEC and LUAD. Compared to LUAD components, SCLC and LCNEC tumors, both of which are high-

grade NEC, demonstrated upregulation of pathways associated with cell proliferation including G2M, E2F, and MYC consistent with the high proliferative nature of SCLC and LCNEC[44]. Importantly, in the pioneer study comparing LCNEC to other lung cancer subtypes from different patients discussed above, George et al. reported that LCNECs were transcriptionally distinct with LUAD and LUSC but closer to SCLC with cell cycle and mitosis-related pathways upregulated in LCNEC comparing to other lung cancer subtypes[45]. Together with our findings, these results highlighted the similarity of LCNECs with SCLCs and suggest that cell proliferation is indeed an important feature of high-grade NEC of lung. Of particular interest, six of nine downregulated pathways in SCLC in our study were inflammatory/immune pathways in line with reported cold immune microenvironment and inferior response to immunotherapy in SCLC[46]. These results also suggest histology-specific modulation of the tumor microenvironment even within the same tumors with the same genetic background and exposure.

In summary, we sought to provide insights to dissect the molecular basis for the histologic determination by multi-omics analysis of three unique datasets: lung cancers of mixed histology that provided a unique opportunity to identify the molecular features associated with different histologic components in the setting of identical genetic background and exposure history; CCLE cell lines of different histology allowing analyzing pure epithelial cancer cells without confounding effect from stromal components; and large cohorts of human lung cancers of different histologic subtypes. Our analysis demonstrated that the different histologic components from the same patients share the majority of point mutations, SCNA, and cancer gene alterations suggesting a shared cell of origin and indicating that histology may not be determined at the genomic level in the majority of tumors. On the other hand, although essentially no genomic mutations were shared, different tumor regions of the same histology across different patients tended to be more closely clustered based on transcriptomic profiles highlighting the presence of histology-specific transcriptomic alterations. It is important to note that tumors of mixed histology are unique biological entities; therefore, different histologic components within these tumors may be different from tumors of pure histology. For example, in our cohort, canonical oncodriver mutations were identified in three of the six tumors with an adenocarcinoma component (SOS1 in Pa34, EGFR/PIK3CA in Pa35, and KRAS/PIK3CA in Pa37) compared to pure LUADs, the majority of which harbor driver mutations. Another major histologic component in our cohort is LCNEC, an aggressive cancer characterized by high proliferation rate and poor prognosis[47,48]. George et al. reported two molecular subtypes of LCNECs based on genomic alterations including "Type I LCNECs" with TP53 and SKT11/KEAP1 alterations and "Type II LCNECs" with inactivation of TP53 and RB1[45]. In the six tumors with LCNEC component in our cohort, one tumor (Pa29: LCNEC mixed with LUSC and NSCLC-NOS) had current TP53/RB1 alterations (TP53 mutation and RB1 LOH); two tumors had concurrent RB1 LOH/STK11 loss (Pa26: LCNEC mixed with LUAD, Pa33: LCNEC mixed with LUSC); one tumor (Pa30: LCNEC mixed with LUAD) had concurrent TP53 mutation/RB1 LOH/STK11 loss; one tumor (Pa34: LCNEC mixed with LUAD) had only STK11 mutation; and one tumor (Pa37: LCNEC mixed with LUAD) had no alterations in TP53, STK11, or RB1 (Fig. 3c, Supplementary Fig. 4, and Supplementary Data 3 and 4) suggesting the biologic features of these LCNEC components from the tumors of mixed histology may not be the same as pure LCNECs. Another major limitation of the current study is the small sample size of tumors of mixed histology. This was due to our intention to focus on tumors of mixed histology with high-grade NEC component. However, mixed tumors that are resected

with a component of LCNEC or SCLC are extremely rare. As such, we analyzed published datasets with NEC included (CCLE, pure epithelial cell components, Karlson et al. and Bhattacharjee et al., larger cohorts but not mixed histology) and focused on the overlap pathways across different datasets. These data suggested that it is possible that histology of lung cancers may be determined at the transcriptomic level, although the exact mechanisms of gene expression regulation remain to be determined. An alternative interpretation, however, is that there is a common mechanistic factor that is driving both histology determination and transcriptomic changes. These intriguing findings warrant validation on larger cohorts of resected tumors of mixed histology harboring NEC components that may require multi-constitutional collaborations and by functional analyses in future studies.

## Methods

**Sample collection and processing**. The current research complies with all relevant ethical regulations. MD Anderson Cancer Center approved the study protocol. Sample selection criteria were: (1) tumors of mixed histology with high-grade NEC component including high-grade LCNEC and small cell carcinoma (SCLC). (2) Enough surgical specimen and matched germline DNA available for multi-omics profiling. Patients with mixed histology lung cancer were included in this study after confirmation with two independent pathologists. The IHC markers were performed in all included cases as part of the diagnostic work up for NECs. The diagnostic criteria for LCNEC are non-small cell carcinomas with neuroendocrine morphology that are positive for at least one neuroendocrine marker (synaptophysin, chromogranin, or CD56). These criteria were strictly followed for cases included. For SCLC, the standard practice was followed that the diagnosis of SCLC can be accurately made on morphologic grounds as established by the guidelines[49–51]; IHC is indicated only if morphology is less than optimal. Unstained slides were microdissected after delineating the different regions of histologic components and then extracted for RNA and DNA. A written informed consent that was approved by the internal review board of the University of Texas M D Anderson Cancer Center was obtained. The study was conducted in accordance with the Declaration of Helsinki.

**Whole-exome sequencing**. DNA was extracted using the QIAamp DNA FFPE Tissue Kit (QIAGEN) and the resulting genomic DNA was sheared into 300–400 bp segments and subjected to library preparation for WES using KAPA library prep (Kapa Biosystems) with the Agilent SureSelect Human All Exon V4 kit according to the manufacturer's instructions. Paired-end multiplex sequencing of DNA samples was performed on the Illumina HiSeq 2000 sequencing platform.

**RNA microarray**. In all, 600 ng RNA per sample was submitted and underwent reverse transcription. Single-strand(ss) cDNA was purified using magnetic beads. The fragmented sscDNA was then hybridized to Affymetrix Clariom S human arrays at 45 °C overnight. Stained arrays are scanned to generate intensity data. All reagent kits and arrays were purchased from Thermo Fisher Scientific.

**Somatic mutation calling and overlapping mutations**. The WES raw FASTQ files were aligned using bwa-mem[52]. Somatic mutations were called using mutect[53] and Lancet (two somatic mutation callers) with tumor-normal pairs following GATK best practice (www.broadinstitute.org/gatk/guide/best-practices.php) for duplicate removal, indel realignment, and base recalibration. Lancet[54] was used for SNV and indel calling using localized colored de Bruijn graph. For SNVs, only those that were called by more than one caller or called in more than one sample from the same patient were retained. For all mutations, we recovered the raw allelic counts from the bam file if it occurred in one of the different histologic subtypes from the same patient. The process was implemented as a Snakemake pipeline and can be found at https://gitlab.com/tangming2005/snakemake_DNAseq_pipeline/tree/multiRG. The number of overlapping mutations across all samples was plotted in an UpSet plot[55] and Venn diagrams.

**Clonal architecture analysis and phylogeny inference**. A high-quality list of SNVs was combined from all samples from the same patient and the allelic counts for those positions were obtained using bam-readcount (https://github.com/genome/bam-readcount). Copy number variations and tumor purity were obtained from sequenza[56], and the mutation allelic counts were analyzed with PyClone for clonality analysis[21]. PyClone was run with 10,000 iterations and a burn-in of 1000 as suggested by the authors. To infer phylogenetic trees, mutation data were converted to the binary data with mutations being 1 and wild-type being 0 and fed into Phangorn R package[57]. Tree topologies were estimated by pratchet, and branch lengths were inferred by acctran.

**Mutational signature and spectrum analysis**. Mutation signatures and spectrum analysis were analyzed by Bioconductor package MutationalPatterns[58] with 30 COSMIC signatures following the standard workflow.

**Somatic copy number analysis (SCNA)**. Copy number analysis was carried out using Sequenza[56]. Both copy number and tumor purity were inferred by Sequenza. Since the signal-to-noise ratio of SCNA could be reduced in the samples with lower tumor purity, we obtained purity-adjusted log2 ratios by $\log 2((\text{original copy ratio} - 1) / \text{purity} + 1)$[59]. The segment files were visualized in IGV[60]. We then used the log2 thresholds of log2(4/2) and log2(1/2) to determine whether a gene is gained or lost focusing only on cancer genes that have shown to have copy number changes in the COSMIC database. The matrices of log2 ratio or binarized copy number status for all genes and cancer genes, respectively, across all samples, were clustered using hierarchical clustering and plotted in a heatmap using ComplexHeatmap[61]. LOH status of RB1 was defined if the B value is equal to 0 from the sequenza output with copy number neutral LOH with B value of 0 and A value of 2 (i.e., a genotype of AA) and a copy number loss LOH with B value of 0 and A value of 1.

**In-house microarray and public microarray/RNAseq data analysis**. The in-house clariom.s.human microarray data were analyzed using Bioconductor packages Oligo[62], pd.clariom.s.human, and limma[63] following standard workflow. GSE94601 microarray data were downloaded using GEOquery[64] and analyzed by the limma package. The Bhattacharjee et al. microarray data were downloaded from http://portals.broadinstitute.org/cgi-bin/cancer/publications/view/62 and analyzed using the affy[65] and limma package. The CCLE lung cancer RNAseq count data were downloaded from the Broad CCLE data portal and processed using DESeq2[66]. Gene set enrichment analysis using Hallmark dataset was carried out using fgsea Bioconductor package[67] and the genes are pre-ranked by (signed log2FoldChange) × −log$_{10}$(p value) for all the public datasets. For the in-house microarray data, we computed the fold change between distinct histologies within the same patient and rank the genes by the fold change.

**Reporting summary**. Further information on research design is available in the Nature Research Reporting Summary linked to this article.

## Data availability

The whole-exome sequencing data have been deposited at the European Bioinformatics Institute European Genome–phenome Archive (EGA) (accession number: EGAS00001005140) through controlled access. The BAM files under accession EGAS00001005140 contain all the raw WES data. To protect patient privacy, interested researchers need to apply via data access committee (DAC), which will grant access upon request. Source data are provided with this study. All other data may be found within the main manuscript or Supplementary information or available from the authors upon request. Public microarray datasets were downloaded from GSE94601, and http://portals.broadinstitute.org/cgi-bin/cancer/publications/view/62. CCLE RNAseq data were downloaded from https://sites.broadinstitute.org/ccle/. Microarray data generated in this study are deposited at GEO with accession number GSE188665. The processed ExpressionSet bioconductor objects can be found at https://osf.io/gxc4r/. The expression matrix of the new microarray data is also provided in the Source Data. The code used to generate the figures can be found at https://github.com/crazyhottommy/mixed_histology_lung_cancer[68]. Source Data are provided with this paper.

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

## Acknowledgements

This work was supported by the generous philanthropic contributions to The University of Texas MD Anderson Lung Moon Shot Program, the MD Anderson Cancer Center Support Grant P30 CA016672, the National Cancer Institute of the National Institute of

Health Research Project Grant (R01CA234629-01), the AACR-Johnson & Johnson Lung Cancer Innovation Science Grant (18-90-52-ZHAN), the MD Anderson Physician Scientist Program, TJ Martell Foundation Award, Sabin Family Foundation Award, Duncan Family Institute Cancer Prevention Research Seed Funding Program, the Cancer Prevention and Research Institute of Texas Multi-Investigator Research Award grant (RP160668), the UT Lung Specialized Programs of Research Excellence Grant (P50CA70907), and Cancer Prevention and Research Institute of Texas (CPRIT) grant RP150079. H.A.A. was supported in part by the T32 NIH fellowship. We thank Jinzhen Chen, Rong Yao, Sally. Boyd, Eric Sisson, and Stan Bujnowski for providing excellent support for high-performance cluster (HPC) resource (http://hpcweb.mdanderson.edu/citing.html).

## Author contributions

Jianjun Z., N.K., and P.A.F. designed the study. M.T. and H.A.A. led the overall data analyses. M.V.N., H.A.A., B.S., C.B., and S.V. collected clinical data. M.R., J.F., C.M., A.W., I.I.W., and N.K. performed pathological assessment. C.-W.C., J.V.H., C.B., L.L., and C.G. performed experiments including DNA/RNA extraction, whole-exome sequencing, and microarray. M.T., H.A.A., X.H., S.M.H., Jianhua Z., J.L., X.M., X.S., W.-C.L., and J.J.L. performed bioinformatics and statistics analyses. M.T., H.A.A., M.V.N., A.R., J.W., B.S., S.S., C.C., J.K., D.G., J.V.H., I.I.W., P.A.F., N.K., and Jianjun Z. interpreted the data. M.T., H.A.A., and Jianjun Z. wrote the manuscript. All authors edited the manuscript.

## Competing interests

Jianjun Zhang reports research funding from Merck, Johnson and Johnson, and consultant fees from BMS, Johnson and Johnson, AstraZeneca, Geneplus, OrigMed, Innovent outside the submitted work. J. V. H. reports research funding from AstraZeneca, GlaxoSmithKline, and Spectrum; consultant fees from AstraZeneca, Boehringer Ingelheim, Bristol-Myers Squibb, Catalyst, EMD Serono, Foundation Medicine, Hengrui Therapeutics, Genentech, GSK, Guardant Health, Eli Lilly, Merck, Novartis, Pfizer, Roche, Sanofi, Seattle Genetics, Spectrum, and Takeda; licensing fees from Spectrum. B. S. reports consultant fees from BMS. M. V. N. reports research funding from Mirati, Novartis, Checkmate, Ziopharm, AstraZeneca, Pfizer, and Genentech; consultant fees from Mirati, Merck/MSD. The other authors declare neither financial nor non-financial interests in the submitted work.
