## [Peer review file · Nature Communications]

REVIEWER COMMENTS

Reviewer #1 (Remarks to the Author): Expert in computational lung cancer genomics

In their manuscript, Ming Tang and colleagues analyzed 19 micro-dissected of tumor regions with different histologic lung cancer subtypes across 9 patients with whole exome sequencing and gene expression arrays. They found that there all tumors of the same patient share a substantial amount of mutations and copy number changes, therefore they are not independent tumors but clonally related. However, no common mutation were reported that could explain the different histological subtypes, such that the authors rule out a genetic mechanism leading to a trans-differentiation into the distinct subtypes. The transcriptome analysis, on the other side, revealed shared pathways between the same histologic subtype from different patients. In total, the manuscript is mostly descriptive, where especially the results from the transcriptome analysis lacks some novelty (see below).

Specific comments:

- 1) A discussion of the main driver genes identified in the different lung cancer subtypes (Collisson et al. Nature 2014; Hammerman et al. Nature 2012; George et al. Nature 2015; George et al. Nature Communications 2018) is missing in the manuscript. In particular, broader discussion of RB1 mutations would be required since these have been associated to trans-differentiation of lung adenocarcinoma to SCLC (June-Koo et al. Journal of Clinical Oncology; Niederst et al. Nature Communications 2015; Peifer et al. Nature Genetics 2012).
- 2) Since the different histological subtypes can be distinguished by several immunohistochemistry markers, it is not surprising that they also show distinct gene expression patterns in the different components. The pathways found to be shared between the histological subtypes recapitulate nicely, what has been found earlier in a study that compares LCNEC to other lung cancer subtypes (George et al. Nature Communications 2018). This result is therefore rather confirmatory but lacks novelty.
- 3) The cancer cell fractions of patients: Pa29 and Pa36 almost lie exactly on the diagonal line and show very little scattering. This would mean (if it is not an artifact from the method used to estimate cancer cell fractions) that the clonal composition is exactly the same in the different histological subtypes, which I found very surprising and should be discussed in the paper. Are the copy number changes also identical in these tumors? To exclude that this result is an artifact from the method, other mutational clustering tools could be used to validate this finding.
- 4) For the remaining patients, phylogenetic trees could be reconstructed and discussed along with driver gene mutations in these samples.
- 5) The quality of the figures could substantially be improved.

Reviewer #2 (Remarks to the Author): Expert in lung cancer genomics and subtypes

The manuscript by Tang et al., "The histologic phenotype of lung cancers may be driven by transcriptomic features rather than genomic characteristics", presents an analysis of histologically heterogeneous lung cancers and their molecular correlates. The existence of mixed histology in lung cancers has presented a mystery in terms of mechanisms of cancer pathogenesis. The authors report that transcriptional profiles are well correlated with histology while genomic alterations are not.

This is an interesting observation that is concordant with previous reports on this topic. The authors have assembled a data set with some potential power—the analysis of microdissected regions of distinct histology using simultaneous exome sequencing and transcriptome sequencing. The authors show that the regions of mixed histology have similar clonal mutations, similar mutational signatures, and similar copy number alterations, but different transcriptional profiles.

However, there are several limitations in the current analysis, including statistical power of the data set and some lack of clarity in methods, as well as over-confidence in conclusions.

1. Statistical power/sample numbers. The sample number for this study, 9 cases, seems very small. According to the introduction, roughly 5% of lung cancer cases have mixed histology. And according to MD Anderson's web page, <https://www.mdanderson.org/cancermoonshots/cancer-types/Lung.html>, over 4,000 lung cancers have been molecularly profiled at MD Anderson over the last 3 years. This would suggest 5% of 4,000 cases, or 200 cases readily available to the researchers. This is far from the 9 cases profiled, and suggests a significant opportunity for a larger case study. A larger study would be able to address the question posed by the researchers in a more statistically powered manner.
2. Sample selection. Although the methods state that the cases of mixed histology were confirmed by two pathologists, the approach and methods for sample selection are not defined. How many samples did the authors consider at first? Which were rejected, and why? A flow-chart for sample selection would be very valuable.
3. Histological analysis. The authors show some examples of mixed histology in Supplementary Figure 1 but this will require more detail. First, the areas selected and dissected of different histology should be shown for each tumor in the supplementary figures. Second, there is no mention of immunohistochemistry (e.g. surfactants or TTF1, keratins, chromogranins, etc.) in the text; I would imagine that IHC is a major method to determine histological subtype in this circumstance, but this is not shown here.
4. It is not entirely clear what is the method for elimination of potential germline contamination in the determination of somatic mutations. For example, in Pa35, there is a 12% concordance of mutations between regions. This implies either a very early divergence or the possibility that many of the common mutations are in fact germline mutations or artifacts that have still passed filtration.
5. For somatic mutations, each type of lung cancer has pathognomonic patterns of mutation. E.g. SCLC has TP53 and RB1 mutations; LUAD has TP53 mutation, EGFR or KRAS or other RTK pathway mutations, and often SMARCA4 or STK11 or KEAP1 mutations; LUSC has TP53 mutations, CDKN2A or RB1 mutations, NFE2L2 or KEAP1 mutations, and PIK3CA or PTEN mutations, for example. LCNEC appears to fall into two classes, one more like LUAD and one more like SCLC. Many of the cancers with mixed histology in this study have LCNEC as one of the histologies, but the authors don't specify the particular class of mutations.

There are also some minor issues, as follows.

- a. Lines 112 through 118, starting with "On the other hand, the same histological subtypes across different patients barely shared any mutations", are not meaningful in my opinion. Different cancers from different patients would not be expected to share specific mutations.
- b. The authors' conclusions on transcriptome driving the cancer phenotype may be overblown. They write in the discussion, for example, that "These data suggested that histology of lung cancers may be determined at the transcriptomic level...". An alternative interpretation is that there is a common mechanistic factor that is driving both histology and transcriptome as two related phenotypes.

Point-by-point Response

Reviewer #1 (Remarks to the Author): Expert in computational lung cancer genomics

In their manuscript, Ming Tang and colleagues analyzed 19 micro-dissected of tumor regions with different histologic lung cancer subtypes across 9 patients with whole exome sequencing and gene expression arrays. They found that there all tumors of the same patient share a substantial amount of mutations and copy number changes, therefore they are not independent tumors but clonally related. However, no common mutation were reported that could explain the different histological subtypes, such that the authors rule out a genetic mechanism leading to a trans-differentiation into the distinct subtypes. The transcriptome analysis, on the other side, revealed shared pathways between the same histologic subtype from different patients. In total, the manuscript is mostly descriptive, where especially the results from the transcriptome analysis lacks some novelty (see below).

Specific comments:

1) A discussion of the main driver genes identified in the different lung cancer subtypes (Collisson et al. Nature 2014; Hammerman et al. Nature 2012; George et al. Nature 2015; George et al. Nature Communications 2018) is missing in the manuscript. In particular, broader discussion of RB1 mutations would be required since these have been associated to trans-differentiation of lung adenocarcinoma to SCLC (June-Koo et al. Journal of Clinical Oncology; Niederst et al. Nature Communications 2015; Peifer et al. Nature Genetics 2012).

Author's Response: We thank the reviewer for this constructive advice and we further agree with the reviewer regarding the relevance of main driver genes in different lung cancer subtypes, particularly RB1 as the main focus of our cohort is neuroendocrine carcinomas. As suggested, we expanded the discussion to include the previously reported relevant driver genes, which now reads as the following. We also cited these milestone papers suggested by the reviewer.

“Cancer gene mutations are known to determine distinct molecular subsets of lung cancers with unique clinical presentation and cancer biology and certain cancer gene mutations are even considered pathognomonic for certain histologic subtypes. Among lung cancers, for example, alterations in EGFR, KRAS, SMARCA4, STK11, KEAP1 etc. are almost exclusively observed in LUADs²⁴; LUSCs often carry mutations in TP53, CDKN2A, RB1, NFE2L2, KEAP1, PIK3CA, PTEN etc.²⁵; 26while RB1 and TP53 are frequently lost in SCLC.” (Line 176-181)

“Furthermore, it has been reported that 1-4% of EGFR-mutant LUADs may transform into SCLCs as one important mechanism underlying drug resistance to EGFR tyrosine kinase inhibitor (TKI) treatment and transformed SCLCs share similar genomic profiles of their parental LUADs^{43,44}. For example, June-Koo et al showed that transformed SCLCs share a common clonal origin with their parental LUADs and complete inactivation of both RB1 and TP53, a genomic hallmark for SCLC, was observed in the original LUADs⁴⁴. Similarly, Niederst et al⁴³ also demonstrated RB1 loss in 100% of transformed SCLCs as well as the original EGFR-mutant LUADs.” (Line 328-334)

2) Since the different histological subtypes can be distinguished by several immunohistochemistry markers, it is not surprising that they also show distinct gene expression patterns in the different components. The pathways found to be shared between the histological subtypes recapitulate nicely, what has been found earlier in a study that compares LCNEC to other lung cancer

subtypes (George et al. Nature Communications 2018). This result is therefore rather confirmatory but lacks novelty.

Author's Response: The authors thank the reviewer for bringing up this important study that is relevant to our current work. In the George et al. study, two subtypes of LCNEC were identified based on genomic alterations: "type I LCNECs" with bi-allelic *TP53* and *SKT11/KEAP1* alterations and "type II LCNECs" with bi-allelic inactivation of *TP53* and *RB1*. Although they share genomic alterations with LUAD and LUSC, both subtypes are transcriptionally distinct from LUAD or LUSC although type I LCNEC is transcriptionally closer to SCLC as revealed by unsupervised consensus clustering. In this pioneer study, LCNEC was compared to other lung cancer subtypes from different patients, where genetic background and environmental exposure history in different patients may confound these findings. The strength of our study is that we analyzed tumors of mixed histology, where different histologic subtypes were from the same tumor/patient sample, which allowed for decoupling of any genetic background noise and exposure that may affect the results. Many of our findings in the current study are consistent with the results from the study by George et al (2018). For example, both George et al and we showed that LCNECs bear up-regulation of pathways and genes controlling cell cycle and mitosis (E2F_target, G2M_checkpoint). Additionally, we also found SCLCs had similar up-regulated pathways compared with LUAD highlighting the transcriptional similarity of LCNEC and SCLC. Our results, together with the findings by George et al, suggest that cell proliferation is indeed an important feature of high-grade neuroendocrine carcinoma of lung. We once again thank the reviewer for bringing our attention to this important work and we believe discussing these key discoveries by George et al. in the context strengthen the findings in our study. Therefore, we cited and discussed this important work, which reads as the following.

"Importantly, in the pioneer study comparing LCNEC to other lung cancer subtypes from different patients discussed above, George et al also reported that LCNECs were transcriptionally distinct with LUAD and LUSC but closer to SCLC with cell cycle and mitosis related pathways up-regulated in LCNEC comparing to other lung cancer subtypes⁴⁴. Together with our findings, these results highlighted the similarity of LCNECs with SCLCs and suggest that cell proliferation is indeed an important feature of high-grade NEC of lung." (Line 355-361)

3) The cancer cell fractions of patients: Pa29 and Pa36 almost lie exactly on the diagonal line and show very little scattering. This would mean (if it is not an artifact from the method used to estimate cancer cell fractions) that the clonal composition is exactly the same in the different histological subtypes, which I found very surprising and should be discussed in the paper. Are the copy number changes also identical in these tumors? To exclude that this result is an artifact from the method, other mutational clustering tools could be used to validate this finding.

Author Response: We thank the reviewer for this critical comment. The reviewer has brought a very important issue for genomic analysis: how to accurately assess the clonal architecture considering difference in copy number aberrations. That was why we utilized PyClone to derive the cancer cell fractions, which is a well-established method for subclonal analysis adjusting for copy number changes and tumor purity. Different histological components in Pa29 and Pa36 are closer to each other in terms of the number of overlapping mutations. The proportion of overlapping mutations between different histologic components for Pa29 and Pa36 was over 90% (Supplementary Fig3), while the copy number profiles between different histologic components are similar albeit not identical (Fig. 3a and Fig 3b). If we plot the raw allele frequency (AF) for each mutation (Reviewer Fig 1), these mutations' raw AFs are overall very similar with each other but do not fall exactly on the diagonal line, which could be due to different local copy number profiles in different histologic components as the reviewer suspected. However, the CCF derived from PyClone showed the subclonal architecture was almost identical between different histologic

components within the same tumors suggesting that PyClone was able to adjust for copy number changes and tumor purity.

Reviewer Fig 1: Scatter plot of raw mutation allele frequency for different histologic components in Pa29 and Pa36. Mutations were labeled with the same color if they were from the same PyClone cluster.

Furthermore, as the reviewer suggested, we performed subclonal analysis using SciClone (Miller et al. 2014). Note that, SciClone uses only mutations located in copy-number neutral, LOH free regions and remove small clusters. Moreover, SciClone does not give the copy-number and tumor purity adjusted allele frequencies. Nevertheless, the subclonal architecture is similar between the different histologic components within the same patient as shown in **Reviewer Fig 2**.

Reviewer Fig 2: Scatter plot of raw mutation allele frequency for different histologic components in Pa29 and Pa36. Mutations were labeled with the same color if they were from the same SciClone cluster.

These results suggest that the subclonal architecture is indeed very similar between different histologic components from the same tumors. One plausible explanation is that the septation of different histologic clones was molecularly late events during evolution of these tumors of mixed histology when the subclonal architecture was already determined and no major genomic evolution has occurred after the separation of different subclones giving rise to different histologic components. We thank the reviewer for bringing up this important point and advised, we discussed these findings in the revised manuscript, which now reads as the following.

“Particularly, Pa29 and Pa36 have the CCFs lined up almost on the diagonal line indicating nearly identical subclonal architecture between different histologic components within the same tumors.....One plausible explanation is that the septation of different histologic clones was molecularly late events during evolution of

these tumors when the subclonal architecture was already determined and no major genomic evolution has occurred after the separation of different subclones giving rise to different histologic components. (Line 163-172)

4) For the remaining patients, phylogenetic trees could be reconstructed and discussed along with driver gene mutations in these samples.

Author Response: Thank you for this constructive suggestion and we agree that phylogenetic analysis can provide novel insight from an evolution view. We therefore constructed Phylogenetic trees of each tumor with driver gene mutation annotated (**Reviewer Fig 3, now new Supplementary Fig 4** in the revised manuscript).

Supplementary Fig. 4

Reviewer Fig 3 (new Supplementary Fig 4). (a-i) Phylogenetics trees of different histologic components within the same tumors. Trunk and branch lengths are proportional to the number of somatic mutations shared between (trunk) or private to different histologic components.

We described these new results as the following.

*“To further delineate the evolution of these tumors of mixed histology and understand the timing of cancer gene mutations, we constructed phylogenetic trees and mapped the canonical cancer gene mutations to the trunks (representing early clonal events before separation of different histologic subclones) and branches (representing later subclonal events after separation of cancer cell subclones that gave rise to different histologic components). As demonstrated in **Supplementary Fig. 4**, the majority of canonical cancer gene mutations were early clonal events before the divergence of different histologic subclones with the exception of PI3KCA, where distinct mutations were identified in different histologic components (P35 and P37).”(Line 198-205)*

5) The quality of the figures could substantially be improved.

Author Response: We apologize for the low quality of some figures. As advised, we re-did all figures to make sure the text font and text size are consistent and make sure they are in high resolution to deliver the information clearly.

The authors thank the reviewer again for these valuable comments and constructive advice. We believe these revisions in response to the reviewer’s comments have significantly improved our manuscript.

Reviewer #2 (Remarks to the Author): Expert in lung cancer genomics and subtypes

The manuscript by Tang et al., “The histologic phenotype of lung cancers may be driven by transcriptomic features rather than genomic characteristics”, presents an analysis of histologically heterogeneous lung cancers and their molecular correlates. The existence of mixed histology in lung cancers has presented a mystery in terms of mechanisms of cancer pathogenesis. The authors report that transcriptional profiles are well correlated with histology while genomic alterations are not.

This is an interesting observation that is concordant with previous reports on this topic. The authors have assembled a data set with some potential power—the analysis of microdissected regions of distinct histology using simultaneous exome sequencing and transcriptome sequencing. The authors show that the regions of mixed histology have similar clonal mutations, similar mutational signatures, and similar copy number alterations, but different transcriptional profiles.

However, there are several limitations in the current analysis, including statistical power of the data set and some lack of clarity in methods, as well as over-confidence in conclusions.

1). Statistical power/sample numbers. The sample number for this study, 9 cases, seems very small. According to the introduction, roughly 5% of lung cancer cases have mixed histology. And according to MD Anderson’s web page, <https://www.mdanderson.org/cancermoonshots/cancer-types/Lung.html>, over 4,000 lung cancers have been molecularly profiled at MD Anderson over the last 3 years. This would suggest 5% of 4,000 cases, or 200 cases readily available to the researchers. This is far from the 9 cases profiled, and suggests a significant opportunity for a larger case study. A larger study would be able to address the question posed by the researchers in a more statistically powered manner.

Author’s Response: We thank the reviewer for this critical question and we further agree that small size was a major limitation of our study. We have put tremendous efforts to identify appropriate tumors for this study, unfortunately these 9 tumors are all we have identified. This was related to the special focus and design of the current study. 1) In this study, we intended to focus on tumors of mixed histology with high-grade neuroendocrine carcinoma (NEC) component including high-grade large cell neuroendocrine carcinoma (LCNEC) and small cell carcinoma (SCLC) because these tumors have very aggressive biology and poor clinical outcome. On the other hand, adenosquamous carcinomas, the most common subtype of lung cancers of mixed histology has been previously studied by several groups. 2) We planned for multi-omics analysis on different histologic components within the same tumors, which requires surgical specimen and matched germline DNA. Obviously, SCLC rarely undergo surgical resection and LCNEC, accounting for 3% of lung cancer, are often non-resectable. Therefore, mixed tumors that are resected with a component of these tumors are extremely rare. We have searched the surgically resected tumors of mixed histology with a component of high-grade NEC that had adequate archival tissue available over the past 10 years and ended up having only 19 tumors in 9 patients that meet our criteria. We are sorry that we were not clear about the study design upfront during the initial submission. We have now revised the manuscript to make this clear in the methods.

*“Sample selection criteria: 1) Tumors of mixed histology with high-grade neuroendocrine carcinoma (NEC) component including high-grade large cell neuroendocrine carcinoma (LCNEC) and small cell carcinoma (SCLC). 2) Enough surgical specimen and matched germline DNA available for multi-omics profiling.”
(Line 396-399)*

With the small sample size fully acknowledged, we analyzed published datasets with NEC included (CCLE, pure epithelial cell components, Karlson et al and Bhattacharjee et al, larger cohorts but not mixed histology) and focused on the overlap pathways across different datasets. With these datasets, we hope that the unique model of paring the RNA and DNA profiles from cancers of mixed histologies in these 9 patients, supported by the bulk RNA confirmatory studies from cell lines and independent cohorts would provide some novel insights regarding biology of lung cancers of mixed histology.

Nevertheless, we agree with the reviewer that larger cohorts of resected tumors of mixed histology including NEC components are warranted in future studies, which will require multi-constititutional collaborations to get enough tumors that meet these criteria. We highlighted this important limitation of our study, which now reads as the following.

*“One major limitation of the current study is the small sample size of tumors of mixed histology. This was due to our intention to focus on tumors of mixed histology with high-grade NEC component. However, mixed tumors that are resected with a component of LCNEC or SCLC are extremely rare. As such, we analyzed published datasets with NEC included (CCLE, pure epithelial cell components, Karlson et al and Bhattacharjee et al, larger cohorts but not mixed histology) and focused on the overlap pathways across different datasets..... These intriguing findings warrant validation on larger cohorts of resected tumors of mixed histology harboring NEC components that may require multi-constititutional collaborations and by functional analyses in future studies.”
(Line378-390)*

2). Sample selection. Although the methods state that the cases of mixed histology were confirmed by two pathologists, the approach and methods for sample selection are not defined. How many samples did the authors consider at first? Which were rejected, and why? A flow-chart for sample selection would be very valuable.

Author’s Response: As mentioned above, we searched all lung cancers resected at MDACC with available archival tissue. The cases included in this study are surgically resected tumors with a component of high-grade NEC that had adequate archival tissue available. We included all tumors that meet these criteria that we could find. We added the sample selection in the Methods section which now reads as the following:

*“Sample selection criteria: 1) Tumors of mixed histology with high-grade neuroendocrine carcinoma (NEC) component including high-grade large cell neuroendocrine carcinoma (LCNEC) and small cell carcinoma (SCLC). 2) Enough surgical specimen and matched germline DNA available for multi-omics profiling.”
(Line 396-399)*

3). Histological analysis. The authors show some examples of mixed histology in Supplementary Figure 1 but this will require more detail. First, the areas selected and dissected of different histology should be shown for each tumor in the supplementary figures. Second, there is no mention of immunohistochemistry (e.g. surfactants or TTF1, keratins, chromogranins, etc.) in the text; I would imagine that IHC is a major method to determine histological subtype in this circumstance, but this is not shown here.

Author’s Response: We thank the reviewer for this critical question. The IHC markers were performed as part of the diagnostic work up for NECs. The diagnostic criteria for LCNEC are non-small cell carcinomas with neuroendocrine morphology that are positive for at least one

neuroendocrine marker (synaptophysin, chromogranin or CD56). These criteria were strictly followed for cases included. For SCLC, the standard practice was followed which is the diagnosis of SCLC can be accurately made on morphologic grounds as established by the guidelines (Travis 2014; Nicholson et al. 2002; Travis et al. 2015).

We thank the reviewer again for bringing this up and we realize that it is important to include these technical details for future readers to understand the data in the context. We therefore revised the methods section (line 380-387) and included representative IHC figures (**Reviewer Fig 4**, new **Supplementary Fig 2**) in the revised manuscript.

“The IHC markers were performed in all included cases as part of the diagnostic work up for NECs. The diagnostic criteria for LCNEC are non-small cell carcinomas with neuroendocrine morphology that are positive for at least one neuroendocrine marker (synaptophysin, chromogranin or CD56). These criteria were strictly followed for cases included. For SCLC, the standard practice was followed which is the diagnosis of SCLC can be accurately made on morphologic grounds as established by the guidelines⁴⁶⁻⁴⁸; IHC is indicated only if morphology is less than optimal.” (Line 400-406)

Supplementary Fig. 2

a

b

c

d

Reviewer Fig 4 (new Supplementary Fig 2). Representative immunohistochemical stains corresponding to the LCNEC component of Pa30 (a) Synaptophysin (b) Chromogranin (c) CD56 (d) TTF-1

4). It is not entirely clear what is the method for elimination of potential germline contamination in the determination of somatic mutations. For example, in Pa35, there is a 12% concordance of mutations between regions. This implies either a very early divergence or the possibility that many of the common mutations are in fact germline mutations or artifacts that have still passed filtration.

Author's Response: We thank the reviewer for this important question. Germline mutation “contamination” is a major confounding factor for cancer genomic studies. This applies to majority of studies using NGS without matched germline DNA control. Fortunately, in our study, we have

matched germline DNA control for each patient (as part of inclusion criteria), which allowed us to specifically call somatic mutations. To increase the accuracy, we specifically used two independent SOMATIC mutation callers (Mutect2 and Lancet) and only mutations identified by the two callers were included. The integration of using two different mutation callers and using matched germline DNA to exclude germline mutations increase the confidence in the mutation calling being somatic in nature. We are sorry that we were not clear with this approach in the initial submission. We have revised the method section to make this clear.

“The whole-exome sequencing raw FASTQ files were aligned using bwa-mem. Somatic mutations were called using mutect and Lancet (two somatic mutation callers) with tumor-normal pairs following GATK best practice...For SNVs, only those which were called by more than one caller or called in more than one sample from the same patient were retained” (Line 420-426)

As the reviewer correctly pointed out, Pa35 has fewer shared mutations (**Fig 1 and Supplementary Fig 3g**) compared with other patients and we agree with the reviewer that this suggests an early divergence for Pa35.

5). For somatic mutations, each type of lung cancer has pathognomonic patterns of mutation. E.g. SCLC has TP53 and RB1 mutations; LUAD has TP53 mutation, EGFR or KRAS or other RTK pathway mutations, and often SMARCA4 or STK11 or KEAP1 mutations; LUSC has TP53 mutations, CDKN2A or RB1 mutations, NFE2L2 or KEAP1 mutations, and PIK3CA or PTEN mutations, for example. LCNEC appears to fall into two classes, one more like LUAD and one more like SCLC. Many of the cancers with mixed histology in this study have LCNEC as one of the histologies, but the authors don't specify the particular class of mutations.

Author's Response: We thank the reviewer for this constructive advice, and we agree that categorizing pathognomonic mutations will make it easier for future readers to follow the story. As such, we revised relevant parts as the following:

“Cancer gene mutations are known to determine distinct molecular subsets of lung cancers with unique clinical presentation and cancer biology and certain cancer gene mutations are even considered pathognomonic for certain histologic subtypes²⁴. Among lung cancers, for example, alterations in EGFR, KRAS SMARCA4, STK11, KEAP1 etc. are almost exclusively observed in LUADs²⁴; LUSCs often carry mutations in TP53, CDKN2A, RB1, NFE2L2, KEAP1, PIK3CA, PTEN etc.²⁵; while RB1 and TP53 are frequently lost in SCLC²⁶... A total of 11 canonical cancer gene mutations, defined as nonsynonymous mutations identical to those previously reported in known cancer genes^{27,28} or truncating mutations in known tumor suppressor genes, were identified in these 19 specimens including mutations of TP53 in the LUSC, LCNEC, SCLC, LUAD components, STK11 and SOS1 in the LCNEC and LUAD components, PIK3CA in the SCLC, LUAD, LUSC and the LCNEC components, and KRAS in the LUAD and LCNEC components.” (line 176-188)

Furthermore, we agree with the reviewer that it is worth discussing the two molecular classes of LCNEC as LCNEC is one major histologic component in our study.

“LCNECs are aggressive lung cancers characterized by high proliferation rate and poor prognosis⁴¹. The genomic features of LCNECs may be associated with different therapeutic response and patient survival⁴². In a pioneer study, George et al reported two molecular subtypes of LCNECs based on genomic alterations including “Type I LCNECs”

with TP53 and SKT11/KEAP1 alterations and “Type II LCNECs” with inactivation of TP53 and RB1. Six of 9 tumors in our cohort had LCNEC component. Interestingly, only P29 (LCNEC mixed with LUSC and NSCLC-NOS) had genomic alterations in TP53 but not in RB1, while the remaining 5 (4 mixed with LUAD and 1 mixed with LUSC) demonstrated genomic alterations in STK11 or KRAS, which was shared between LCNEC and counterpart histologic components within the same tumors in all 5 patients (Fig 4b,c,h,k and Supplementary Fig 4).” (Line 317-326)

There are also some minor issues, as follows.

a). Lines 112 through 118, starting with “On the other hand, the same histological subtypes across different patients barely shared any mutations”, are not meaningful in my opinion. Different cancers from different patients would not be expected to share specific mutations.

Author’s Response: We thank the reviewer for this critical comment and we removed this sentence in the revised manuscript.

b). The authors’ conclusions on transcriptome driving the cancer phenotype may be overblown. They write in the discussion, for example, that “These data suggested that histology of lung cancers may be determined at the transcriptomic level...”. An alternative interpretation is that there is a common mechanistic factor that is driving both histology and transcriptome as two related phenotypes.

Author’s Response: The authors thank the reviewer for this critical comment and we agree that this was an overstatement from our data. We therefore rephrased the sentence and discussed the alternative interpretation, which now reads as the following.

“These data suggested it is possible that histology of lung cancers may be determined at the transcriptomic level although the exact mechanisms of gene expression regulation remain to be determined. An alternative interpretation, however, is that there is a common mechanistic factor that is driving both histology determination and transcriptomic changes.” (line 384-388)

Accordingly, we revised the abstract as the following.

“These data derived from mixed histologic subtypes in the setting of identical genetic background and exposure history support that histologic fate of lung cancer cells is associated with transcriptomic features rather than the genomic profiles.”

Furthermore, we changed the title to “The histologic phenotype of lung cancers **is associated with** transcriptomic features rather than genomic characteristics”.

The authors thank the reviewer again for these valuable comments and constructive advice. We believe these revisions in response to the reviewer’s comments have significantly improved our manuscript.

References

- Lee, June-Koo, Junehawk Lee, Sehui Kim, Soyeon Kim, Jeonghwan Youk, Seongyeol Park, Yohan An, et al. 2017. "Clonal History and Genetic Predictors of Transformation into Small-Cell Carcinomas from Lung Adenocarcinomas." *Journal of Clinical Oncology: Official Journal of the American Society of Clinical Oncology* 35 (26): 3065–74.
- Miller, Christopher A., Brian S. White, Nathan D. Dees, Malachi Griffith, John S. Welch, Obi L. Griffith, Ravi Vij, et al. 2014. "SciClone: Inferring Clonal Architecture and Tracking the Spatial and Temporal Patterns of Tumor Evolution." *PLoS Computational Biology* 10 (8): e1003665.
- Nicholson, Siobhan A., Mary Beth Beasley, Elizabeth Brambilla, Philip S. Hasleton, Thomas V. Colby, Mary N. Sheppard, Roni Falk, and William D. Travis. 2002. "Small Cell Lung Carcinoma (SCLC): A Clinicopathologic Study of 100 Cases with Surgical Specimens." *The American Journal of Surgical Pathology* 26 (9): 1184–97.
- Niederst, Matthew J., Lecia V. Sequist, John T. Poirier, Craig H. Mermel, Elizabeth L. Lockerman, Angel R. Garcia, Ryohei Katayama, et al. 2015. "RB Loss in Resistant EGFR Mutant Lung Adenocarcinomas That Transform to Small-Cell Lung Cancer." *Nature Communications* 6 (1): 6377.
- Peifer, Martin, Lynnette Fernández-Cuesta, Martin L. Sos, Julie George, Danila Seidel, Lawryn H. Kasper, Dennis Plenker, et al. 2012. "Integrative Genome Analyses Identify Key Somatic Driver Mutations of Small-Cell Lung Cancer." *Nature Genetics* 44 (10): 1104–10.
- Travis, William D. 2014. "Pathology and Diagnosis of Neuroendocrine Tumors: Lung Neuroendocrine." *Thoracic Surgery Clinics* 24 (3): 257–66.
- Travis, William D., Elisabeth Brambilla, Andrew G. Nicholson, Yasushi Yatabe, John H. M. Austin, Mary Beth Beasley, Lucian R. Chirieac, et al. 2015. "The 2015 World Health Organization Classification of Lung Tumors: Impact of Genetic, Clinical and Radiologic Advances since the 2004 Classification." *Journal of Thoracic Oncology: Official Publication of the International Association for the Study of Lung Cancer* 10 (9): 1243–60.

REVIEWER COMMENTS

Reviewer #1 (Remarks to the Author):

The authors have adequately addressed all my concerns, I have no further comments.

Reviewer #2 (Remarks to the Author):

The revised manuscript is much improved based on the suggestions from this reviewer and others. The revised discussion of the histology is a significant improvement as is the inclusion of genomic data.

I remain confused on one point. The authors describe the somatic mutations in the samples but do not show them in detail. A supplementary table or figure would be helpful. My specific confusion is as follows: the authors describe a total of 11 somatic driver mutations in 19 specimens. As I understand it, the typical NSCLC (or SCLC) has on the order of 2 detectable driver mutations, or more, per case; this would give an expected number of 38 driver mutations in the 19 specimens.

How do the authors account for the low number of somatic driver mutations? One possible explanation is that they are not counting repeats--that would be clarified by a table of driver mutations per specimen. Another possible interpretation would be that the cancer cell fraction is low in these specimens and that this is limiting detection; it would be important for the authors to address this question by describing the mutation fraction in the suggested table and thereby to provide an estimate of cancer cell fraction.

Finally, in this reviewers' opinion, the study is interesting but is perhaps more suitable for a journal such as Scientific Reports than for Nature Communications.

Reviewer #3 (Remarks to the Author): Expert in lung histopathology

The authors have satisfactorily responded to the prior reviewers' comments for the most part. I, as an independent reviewer, have a few additional comments/questions.

1. The authors acknowledged the small sample size as the limitation of the study, but the unique nature of the cohort should also be noted since only one of the six tumors with an adenocarcinoma component harbored a driver mutation (KRAS) (opposed to the majority of lung adenocarcinomas harboring driver mutations).

2. While the authors emphasized the difference in transcriptomic profiles between small cell and adenocarcinoma histology with tumors 31 and 35, but they are those that showed the least fractions of shared mutations among the tumors studied, and Rb loss does not appear to be shared between the two histology components in either of them; thus, the contribution of genetic diversity to the difference in histology cannot be completely excluded.

3. Lines 322-326: Interestingly, only P29 (LCNEC mixed with LUSC and NSCLC-NOS) had genomic alterations in TP53 but not in RB1 while the remaining 5 (4 mixed with LUAD and 1 mixed with LUSC) demonstrated genomic alterations in STK11 or KRAS, which was shared between LCNEC and counterpart histologic components within the same tumors in all 5 patients (Fig. 4b, c, h, k and Supplementary Fig. 4).

Based on the figures, only 2 of the remaining 5 showed shared SKT11 or KRAS mutations, but it does not seem that the others have either TP53, SKT11 or KRAS mutations.

4. Lines 239-242: Two pairs of LCNEC and SCLC components from different patients clustered together in the transcriptome analysis.

Since the differentiation of LCNEC and SCLC can be challenging based on the morphology and immunophenotype alone, I would like to make sure that those SCLC samples harbored concurrent TP53 and Rb alterations.

Minor issues:

1. Line 384: "These data suggested" is redundant.
2. Line 435: The sentence that ends with readout) needs a period.

Point-by-point response

Reviewer #2 (Remarks to the Author):

The revised manuscript is much improved based on the suggestions from this reviewer and others. The revised discussion of the histology is a significant improvement as is the inclusion of genomic data.

Author's Response: We are happy that the reviewer was overall satisfied with our revision.

1. I remain confused on one point. The authors describe the somatic mutations in the samples but do not show them in detail. A supplementary table or figure would be helpful. My specific confusion is as follows: the authors describe a total of 11 somatic driver mutations in 19 specimens. As I understand it, the typical NSCLC (or SCLC) has on the order of 2 detectable driver mutations, or more, per case; this would give an expected number of 38 driver mutations in the 19 specimens.

Author's Response: We thank the reviewer for this constructive advice, and we agree a list of all mutations included in these analyses will be informative. As suggested, we added **Supplementary Data 2** listing all somatic mutations in each sample. Regarding the driver mutations, please see our explanation and revision following the reviewer's **Specific Comment 2**.

2. How do the authors account for the low number of somatic driver mutations? One possible explanation is that they are not counting repeats--that would be clarified by a table of driver mutations per specimen. Another possible interpretation would be that the cancer cell fraction is low in these specimens and that this is limiting detection; it would be important for the authors to address this question by describing the mutation fraction in the suggested table and thereby to provide an estimate of cancer cell fraction. Finally, in this reviewers' opinion, the study is interesting but is perhaps more suitable for a journal such as Scientific Reports than for Nature Communications.

Author's Response: We appreciate that the reviewer brought up this important point and we thank the reviewer's suggestions to make these clear for future readers. There are two reasons for the observed low number of somatic driver mutations: 1) Just as the reviewer suspected, we did not count repeats. We have now added **Supplementary Data 3** to show the driver mutations in each sample as the reviewer suggested. 2) We applied very strict criteria to define cancer gene mutations. Only cancer gene mutations identical to those reported in both COSMIC and CiVIC databases were included in our original list of cancer gene mutations. We now realize that these criteria may be overly strict, leading to missing critical cancer gene mutations. Therefore, in the revised manuscript, we included all identical cancer gene mutations previously reported in COSMIC database. With the new criteria, there were a total of 34 cancer gene mutations detected in these 19 specimens.

As the reviewer pointed out, low tumor purity is a common problem for missing mutations. In our study, tumors were processed under strict quality control and cancer cells were enriched by microdissection for all specimens. Therefore, the tumor purity was overall pretty good in our cohort. Additionally, we inferred cancer cell fraction by PyClone for each mutation as suggested by the reviewer, which is now included in **Supplementary Data 2** and **3** (cellular prevalence column).

We thank the reviewer again for these constructive advices to make these points clear for future readers.

Reviewer #3 (Remarks to the Author): Expert in lung histopathology

The authors have satisfactorily responded to the prior reviewers' comments for the most part. I, as an independent reviewer, have a few additional comments/questions.

Author's Response: We thank the reviewer for the overall favorable comments.

1. The authors acknowledged the small sample size as the limitation of the study, but the unique nature of the cohort should also be noted since only one of the six tumors with an adenocarcinoma component

harbored a driver mutation (KRAS) (opposed to the majority of lung adenocarcinomas harboring driver mutations).

Author's Response: We thank the reviewer for this important point. Lower number of driver mutations is due to our strict definition of the driver mutations: only identical driver mutations reported in both COSMIC and CiVIC databases were included in our original list of cancer gene mutations. We now realized that these criteria may be overly strict, leading to missing critical cancer gene mutations. Therefore, in the revised manuscript, we included all cancer gene mutations previously reported in COSMIC database (**Supplementary Data 3**). With the new criteria, three of the six tumors with adenocarcinoma components had oncdriver mutations identified (**Supplementary Data 3; Supplementary Figure 4**). Having said that, the reviewer made a good point that these adenocarcinoma components from tumors of mixing histology may be different from de novo adenocarcinomas. We discussed this important point as the following.

"It is important to note that tumors of mixed histology are unique biological entities, therefore, different histologic components within these tumors may be different from tumors of pure histology. For example, in our cohort, canonical oncdriver mutations were identified in 3 of the 6 tumors with an adenocarcinoma component (SOS1 in Pa34, EGFR/PIK3CA in Pa35 and KRAS/PIK3CA in Pa37) compared to pure LUADs, the majority of which harbor driver mutations." (line 370 – 377).

2. While the authors emphasized the difference in transcriptomic profiles between small cell and adenocarcinoma histology with tumors 31 and 35, but they are those that showed the least fractions of shared mutations among the tumors studied, and Rb loss does not appear to be shared between the two histology components in either of them; thus, the contribution of genetic diversity to the difference in histology cannot be completely excluded.

Author's Response: The reviewer made a great point. We agree with the reviewer's observation and interpretation. We apologize that we failed to mention that although no RB1 mutations were detected, RB1 LOH was observed in the SCLC and LUAD components from both Pa31 and Pa35. Having said that, with only 12.1% mutations shared between SCLC and LUAD components of Pa35, we cannot exclude the contribution of genetic alterations in histologic determination in some tumors. We therefore revised our manuscript as the following.

"These results are consistent with previous findings from adenosquamous lung cancers, suggesting somatic mutations may not be the primary determinants of histology in most tumors. On the other hand, in Pa35, only 12.1% mutations were shared between the SCLC and LUAD components. Therefore, we cannot exclude the contribution of genetic alterations in histologic determination in a subset of tumors." (line 113-117).

3. Lines 322-326: Interestingly, only P29 (LCNEC mixed with LUSC and NSCLC-NOS) had genomic alterations in TP53 but not in RB1 while the remaining 5 (4 mixed with LUAD and 1 mixed with LUSC) demonstrated genomic alterations in STK11 or KRAS, which was shared between LCNEC and counterpart histologic components within the same tumors in all 5 patients (Fig. 4b, c, h, k and Supplementary Fig. 4). Based on the figures, only 2 of the remaining 5 showed shared SKT11 or KRAS mutations, but it does not seem that the others have either TP53, SKT11 or KRAS mutations.

Author's Response: This was a mistake in the text and the figure was correct. We apologize for this mistake, and we thank the reviewer for catching this. We have corrected this error.

4. Lines 239-242: Two pairs of LCNEC and SCLC components from different patients clustered together in the transcriptome analysis. Since the differentiation of LCNEC and SCLC can be challenging based on the morphology and immunophenotype alone, I would like to make sure that those SCLC samples harbored concurrent TP53 and Rb alterations.

Author's Response: We thank the reviewer for this critical comment. The reviewer is 100% correct that one of the biggest flaws of current criteria for LCNEC is the overlap with SCLC, which makes it difficult to distinguish LCNEC from SCLC for some cases, particularly for small biopsy samples. On the other hand, the distinction is much more accurate in resected specimens. All the tumors included in this study were surgically resected specimens and the diagnosis was confirmed by all thoracic pathologists (M.R., C.M., A.W. and N.K.) without equivocation. Please see the HE images of Pa31T1 and Pa35T1, the two SCLC components that were clustered with the LCNEC (**Reviewer Figure 1**). Furthermore, Pa31T1 harbored a TP53 p.135F and Pa35T1 had a TP53 p.C176Y mutation, respectively and RB1 LOH (**Supplementary Data 4**) was detected in both specimens providing molecular support of their diagnosis as SCLCs.

Reviewer Figure 1. High magnification images of the small cell carcinoma components in patients 31 (left) and 35 (right). The HE slides show sheets of tumor cells with high N:C, scant cytoplasm, and absent or inconspicuous nucleoli, frequent mitotic figures and apoptotic cells, features characteristic of small cell carcinoma.

Minor issues:

1. Line 384: "These data suggested" is redundant.

Author's Response: We apologize for the typo, and we have now removed the redundant phrase. We have also thoroughly proof-read the manuscript for typos and grammar errors.

2. Line 435: The sentence that ends with readout) needs a period.

Author's Response: We thank the reviewer's meticulous review. We have now added the period.

The authors thank the reviewer again for these insightful and constructive suggestions to make our manuscript clearer and stronger.

REVIEWERS' COMMENTS

Reviewer #3 (Remarks to the Author):

The author have responded to the reviewers' comments reasonably well. I don't have any additional suggestions.